# Voltage-dependent dynamics of the BK channel cytosolic gating ring are coupled to the membrane-embedded voltage sensor

Pablo Miranda[1]*, Miguel Holmgren[1], Teresa Giraldez[2,3]*

[1]National Institute of Neurological Disorders and Stroke, National Institutes of Health, Bethesda, United States; [2]Departamento de Ciencias Medicas Basicas, Universidad de La Laguna, San Cristóbal de La Laguna, Spain; [3]Instituto de Tecnologias Biomedicas, Universidad de La Laguna, San Cristóbal de La Laguna, Spain

**Abstract** In humans, large conductance voltage- and calcium-dependent potassium (BK) channels are regulated allosterically by transmembrane voltage and intracellular $Ca^{2+}$. Divalent cation binding sites reside within the gating ring formed by two Regulator of Conductance of Potassium (RCK) domains per subunit. Using patch-clamp fluorometry, we show that $Ca^{2+}$ binding to the RCK1 domain triggers gating ring rearrangements that depend on transmembrane voltage. Because the gating ring is outside the electric field, this voltage sensitivity must originate from coupling to the voltage-dependent channel opening, the voltage sensor or both. Here we demonstrate that alterations of the voltage sensor, either by mutagenesis or regulation by auxiliary subunits, are paralleled by changes in the voltage dependence of the gating ring movements, whereas modifications of the relative open probability are not. These results strongly suggest that conformational changes of RCK1 domains are specifically coupled to the voltage sensor function during allosteric modulation of BK channels.
DOI: https://doi.org/10.7554/eLife.40664.001

*For correspondence:
pablo.mirandafernandez2@nih.
gov (PM);
giraldez@ull.edu.es (TG)

**Competing interests:** The authors declare that no competing interests exist.

## Introduction

The open probability of large conductance voltage-and $Ca^{2+}$-activated $K^+$ (BK or slo1) channels is regulated allosterically by voltage and intracellular concentration of divalent ions (*Barrett et al., 1982*; *Moczydlowski and Latorre, 1983*; *Horrigan and Aldrich, 2002*; *Latorre et al., 2017*). This feature makes BK channels important regulators of physiological processes such as neurotransmission and muscular function, where they couple membrane voltage and the intracellular concentration of $Ca^{2+}$ (*Robitaille and Charlton, 1992*; *Hu et al., 2001*; *Wang et al., 2001*; *Raffaelli et al., 2004*). The BK channel is formed in the membrane as tetramers of α subunits, encoded by the KCNMA1 gene (*Shen et al., 1994*; *Quirk and Reinhart, 2001*). Each α subunit contains seven transmembrane domains (S0 to S6), a small extracellular N-terminal domain and a large intracellular C-terminal domain (*Wallner et al., 1996*; *Meera et al., 1997*; *Tao et al., 2017*) (Figure 2a). Similar to other voltage-gated channels, the voltage across the membrane is sensed by the voltage sensor domain (VSD), containing charged amino acids within transmembrane segments S2, S3 and S4 (*Díaz et al., 1998*; *Ma et al., 2006*; *Pantazis and Olcese, 2012*; *Tao et al., 2017*). The sensor for divalent cations is at the C-terminal region and is formed by two Regulator of Conductance for $K^+$ domains (RCK1 and RCK2) per α subunit (*Wei et al., 1994*; *Moss and Magleby, 2001*; *Xia et al., 2002*; *Zeng et al., 2005*; *Wu et al., 2010*). In the tetramer, four RCK1-RCK2 tandems pack against each

other in a large structure known as the gating ring (*Wu et al., 2010*; *Yuan et al., 2011*; *Giraldez and Rothberg, 2017*; *Tao et al., 2017*; *Zhou et al., 2017*). Two high-affinity $Ca^{2+}$ binding sites are located in the RCK2 (also known as 'Ca$^{2+}$ bowl') and RCK1 domains, respectively. Additionally, a site with low affinity for $Mg^{2+}$ and $Ca^{2+}$ is located at the interface between the VSD and the RCK1 domain (*Shi and Cui, 2001*; *Zhang et al., 2001*; *Bao et al., 2002*; *Xia et al., 2002*; *Yang et al., 2007*; *Yang et al., 2008a*; *Tao et al., 2017*) (Figure 2a). The high-affinity binding sites show structural dissimilarity (*Zhang et al., 2010*; *Tao et al., 2017*) and different affinity for divalent ions (*Zeng et al., 2005*). Apart from $Ca^{2+}$, it has been described that $Cd^{2+}$ selectively binds to the RCK1 site, whereas $Ba^{2+}$ and $Mg^{2+}$ show higher affinity for the RCK2 site (*Xia et al., 2002*; *Zeng et al., 2005*; *Yang et al., 2008b*; *Zhou et al., 2012*; *Miranda et al., 2016*). Thus, intracellular concentrations of $Ca^{2+}$, $Cd^{2+}$, $Ba^{2+}$ or $Mg^{2+}$ can shift the voltage dependence of BK activation towards more negative potentials. Using patch clamp fluorometry (PCF), we have shown that these cations trigger independent conformational changes of RCK1 and/or RCK2 within the gating ring, measured as large changes in the efficiency of Fluorescence Resonance Energy Transfer (FRET) between fluorophores introduced into specific sites in the BK tetramer. These rearrangements depend on the specific interaction of the divalent ions with their high-affinity binding sites, showing different dependences on cation concentration and membrane voltage (*Miranda et al., 2013*; *Miranda et al., 2016*). To date, the proposed transduction mechanism by which divalent ion binding increases channel open probability was a conformational change of the gating ring that leads to a physical pulling of the channel gate, where the linker between the S6 transmembrane domain and the RCK1 region acts like a passive spring (*Niu et al., 2004*). Such a mechanism would be analogous to channel activation by ligand binding in glutamate receptor or cyclic nucleotide-gated ion channels, also tetramers (*Sobolevsky et al., 2009*; *James et al., 2017*). Our previous results do not support this as the sole mechanism underlying coupling of divalent ion binding to channel opening, since the gating ring conformational changes that we have recorded: 1) are not strictly coupled to the opening of the channel's gate, and 2) show different voltage dependence for each divalent ion. In addition, the recent cryo-EM structure of the full slo1 channel of *Aplysia californica* (*Hite et al., 2017*; *Tao et al., 2017*) shows that the RCK1 domain of the gating ring is in contact with the VSD, predicting that changes in the voltage sensor position could be reflected in the voltage dependent gating ring reorganizations.

Understanding the nature of the voltage dependence associated with individual rearrangements produced by binding of divalent ions to the gating ring is essential to untangle the mechanism underlying the role of such rearrangements in BK channel gating. To this end, we have now performed PCF measurements with human BK channels heterologously expressed in *Xenopus* oocytes, including a range of VSD mutations or co-expressed with different regulatory subunits. Here we provide evidence for a functional interaction between the gating ring and the voltage sensor in full-length, functional BK channels at the plasma membrane, in agreement with the structural data from *Aplysia* BK. Moreover, these data support a pathway that couples to divalent ion binding to channel opening through the voltage sensor.

## Results

### Voltage dependence of gating ring rearrangements is associated to activation of the RCK1 binding site

BK α subunits labeled with fluorescent proteins CFP and YFP in the linker between the RCK1 and RCK2 domains (position 667) retain the functional properties of wild-type BK channels (*Miranda et al., 2013*; *Miranda et al., 2016*). This allowed us to use PCF to detect conformational rearrangements of the gating ring measured as changes in FRET efficiency (*E*) between the fluorophores (*Miranda et al., 2013*; *Miranda et al., 2016*). Binding of $Ca^{2+}$ ions to both high-affinity binding sites (RCK1 and $Ca^{2+}$ bowl) produces an activation of BK channels, coincident with an increase in *E* from basal levels reaching saturating values at high $Ca^{2+}$ concentrations (*Miranda et al., 2013* and *Figure 1a*). In addition, we observed that the *E* signal has the remarkable property that in intermediate $Ca^{2+}$ concentrations (from 4 μM to 55 μM) it shows voltage dependence besides its $Ca^{2+}$ dependence (*Miranda et al., 2013* and *Figure 1a*). As discussed previously (*Miranda et al., 2013*), these changes in *E* with voltage are not conformational dynamics of the gating ring that simply follow the

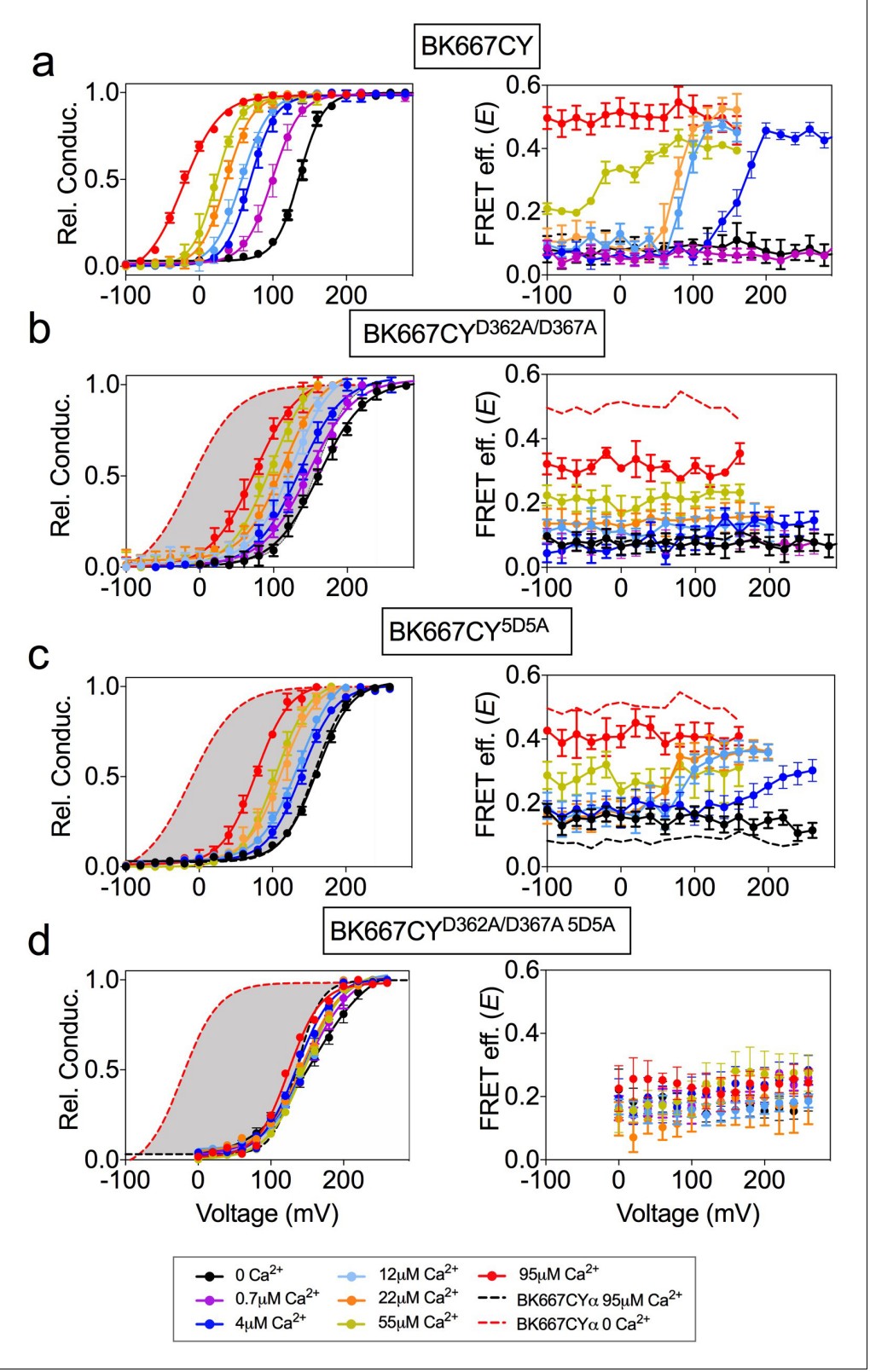

**Figure 1.** Voltage dependence of gating ring rearrangements is associated to activation of the RCK1 binding site. G-V (left panels) and *E*-V curves (right panels) obtained simultaneously at several $Ca^{2+}$ concentrations from (**a**) the BK667CY construct, (**b**) mutation of the RCK1 high-affinity site (D362A/D367A), (**c**) mutation of the $Ca^{2+}$ bowl (5D5A), or (**d**) both (D362A/D367A 5D5A). Note that the voltage dependence of the *E* signal is only abolished after

*Figure 1 continued on next page*

*Figure 1 continued*

mutating the RCK1 high-affinity binding site (**b**) or both (**d**). Data corresponding to each $Ca^{2+}$ concentration are color-coded as indicated in the legend at the bottom. Solid curves in the G-V graphs represent Boltzmann fits. For reference, grey shadows in (**a–d**) left panels represent the full range of G-V curves corresponding to non-mutated BK667CY channels from 0 μM $Ca^{2+}$ to 95 μM $Ca^{2+}$ (indicated with colored dashed lines). Data points and error bars represent average ± SEM (*n* = 3–14, N = 2–8). Part of the data in (**a, b** and **c**) are taken from (*Miranda et al., 2013*) and (*Miranda et al., 2016*).

DOI: https://doi.org/10.7554/eLife.40664.002

voltage dependence of VSD. For instance, at 0 $Ca^{2+}$ concentrations movements of the VSD occurs between 0 and +300 mV (*Stefani et al., 1997*; *Horrigan et al., 1999*; *Horrigan and Aldrich, 2002*; *Zhang et al., 2014*; *Carrasquel-Ursulaez et al., 2015*; *Zhang et al., 2017*). However, we do not observe changes in *E* between 0 and +240 mV (*Figure 1a*). Similarly, at 100 μM $Ca^{2+}$, charge movement takes place between −100 and +150 mV (*Carrasquel-Ursulaez et al., 2015*), while our FRET signals at 95 μM $Ca^{2+}$ do not vary within this voltage range (*Figure 1a*). Independent activation of high-affinity binding sites by other divalent ions ($Ba^{2+}$, $Cd^{2+}$, or $Mg^{2+}$ (*Miranda et al., 2016*)) led us to postulate that $Ca^{2+}$ activation has a site-dependent relation to voltage. To further evaluate the effect of individual high-affinity $Ca^{2+}$ binding sites on the voltage-dependent component of the gating ring conformational changes we first selectively mutated the binding sites. Mutations D362A and D367A (*Xia et al., 2002*; *Zeng et al., 2005*) were introduced in the BK667CY construct (BK667CY$^{D362A/D367A}$) to remove the high-affinity binding site located in the RCK1 domain. *Figure 1b* shows the relative conductance and *E* values for the BK667CY$^{D362A/D367A}$ construct at different membrane voltages for various $Ca^{2+}$ concentrations. As described previously, the G-V curves show a significantly reduced shift to more negative potentials when $Ca^{2+}$ is increased, as compared to the non-mutated BK667CY (*Figure 1a–b*, left panels). Specific activation of the $Ca^{2+}$ bowl renders a smaller change in *E* values, which are not voltage-dependent within the voltage range tested (*Figure 1b*, right panel). To test the effect of eliminating the RCK2 $Ca^{2+}$ binding site -the $Ca^{2+}$ bowl- we mutated five aspartates to alanines (5D5A) (*Bao et al., 2002*). As expected, activation of only the RCK1 domain by $Ca^{2+}$ reduced the $Ca^{2+}$-dependent shift in the GV curves (*Figure 1c*, left panel). Even though the extent to which the *E* values changed with $Ca^{2+}$ was reduced (*Figure 1c*), there was a persistent voltage dependence equivalent to that shown in *Figure 1a* corresponding to the non-mutated channel (most appreciable at 12 μM and 22 μM $Ca^{2+}$ concentrations; *Figure 1c*, right panel) (*Miranda et al., 2013*). Further, at these two $Ca^{2+}$ concentrations the changes in *E* occurred within the same voltage range (+60–120 mV) in channels with the $Ca^{2+}$ bowl mutated (*Figure 1c*) or not (*Figure 1a*). This effect seems not to be attributable to $Ca^{2+}$ binding to unknown binding sites in the channel, since the double mutation of the RCK1 and RCK2 sites abolishes the change in the FRET signal (*Figure 1d*). Altogether, these results indicate that the voltage-dependent component of the gating ring conformational changes triggered by $Ca^{2+}$ in the BK667CY construct depends on activation of the RCK1 binding site. Because the gating ring is not within the transmembrane region, it is not expected to be directly influenced by the transmembrane voltage. Therefore, the voltage-dependent FRET signals must be coupled to the dynamics of the gate region associated with the opening and closing of the channel and/or those of the voltage sensor domain.

## The voltage-dependent conformational changes of the gating ring are not related to the opening and closing of the pore domain

To test whether the voltage-dependent FRET signals relate to the opening and closing of the channel (intrinsic gating) we used two modifications of BK channel function in which the relative probability of opening is shifted in the voltage axis, yet the actual dynamics of voltage sensor are expected to be unaltered (*Figure 2b*). We reasoned that, if the voltage-dependent FRET signals of the gating ring are coupled to the opening and closing, they should follow a similar displacement with voltage. The first BK channel construct is the α subunit including the single point mutation F315A, which has been described to shift the voltage dependence of the relative conductance of the channel to more positive potentials, by uncoupling the voltage sensor activation from the gate opening (*Figure 2c*) (*Carrasquel-Ursulaez et al., 2015*). *Figure 2d* shows the relative conductance and *E* vs. voltage for the BK667CY$^{F315A}$ mutant at various $Ca^{2+}$concentrations. Our results show that the shift of the

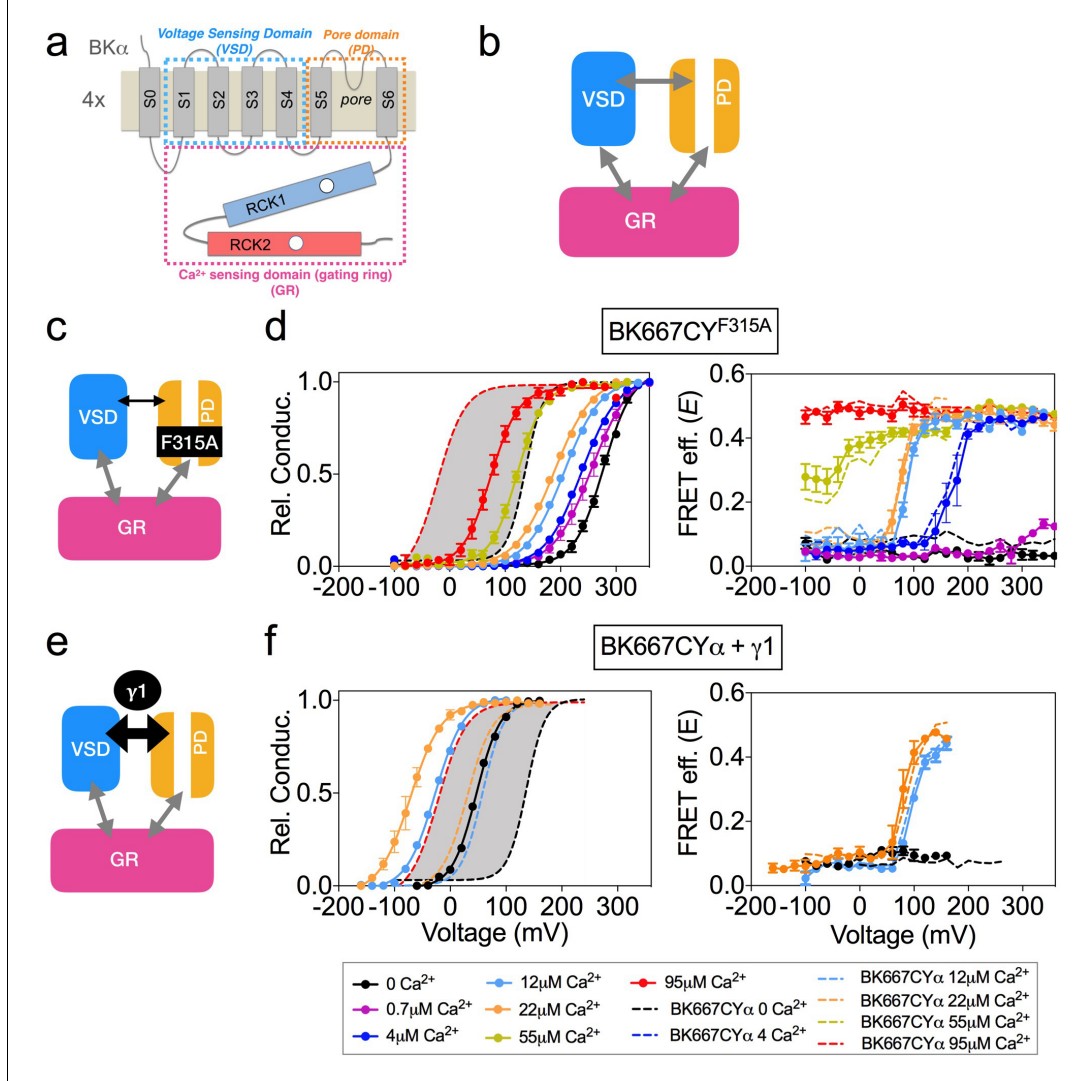

**Figure 2.** Modification of the voltage dependence of gate opening does not affect the gating ring voltage-dependent conformational changes. (**a**) Topology of the BKα subunit where the voltage sensing domain (VSD), $Ca^{2+}$ sensing domain (gating ring, GR) and pore domain (PD) are indicated by colored dashed lines boxes (see main text for a full description). (**b**) The three BK functional modules (VSD, PD, GR), schematically represented as colored boxes, interact allosterically. (**c**) Diagram representing the main effect of the F315A mutation, which is the uncoupling of the VSD to the PD. (**d**) G-V (left panel) and $E$-V curves (right panel) obtained simultaneously at several $Ca^{2+}$ concentrations after mutation of the F315 site to alanine (BK667CY$^{F315A}$). It should be noted that the extent of the shifts induced by the mutation are smaller than previously reported (*Carrasquel-Ursulaez et al., 2015*), which could arise from the different experimental conditions and/or our fluorescent construct. (**e**) The interaction with the γ1 subunit favors the VSD-PD coupling mechanism (**f**) G-V (left) and $E$-V curves (right) of BK667CY α subunits co-expressed with γ1 subunits. In all panels, data corresponding to each $Ca^{2+}$ concentration are color-coded as indicated in the bottom legend. Colored dashed lines represent the G-V and $E$-V curves corresponding to BK667CYα channels (*Miranda et al., 2013*; *Miranda et al., 2016*). The solid curves in the G-V graphs represent Boltzmann fits. The full range of G-V curves from 0 μM $Ca^{2+}$ to 95 μM $Ca^{2+}$ from BK667CY is represented as a grey shadow in left panels (**d** and **f**), for reference. Data points and error bars represent average ± SEM ($n$ = 3–8; N = 2–3).

DOI: https://doi.org/10.7554/eLife.40664.003

relative probability of opening to more positive potentials (*Figure 2d*, left panel) does not lead to changes in the voltage dependence of the gating ring FRET signals (*Figure 2d*, right panel).

The second modification of BK function consisted in co-expressing the wild type α subunit with the auxiliary subunit γ1 (*Yan and Aldrich, 2010*; *Yan and Aldrich, 2012*; *Gonzalez-Perez et al., 2014*; *Li and Yan, 2016*). In this case, the relative probability of opening is shifted to more negative potentials by increasing the coupling between the voltage sensor and the gate of the channel

(*Figure 2e*). This construct adds the advantage of representing a physiologically relevant modification of channel gating. *Figure 2f* shows the relative conductance and *E vs.* voltage in oocytes co-expressing the BK667CYα and γ1 at voltages ranging from −160 to +260 mV, with three [Ca$^{2+}$] concentrations: nominal 0, 12 μM and 22 μM. As expected, the presence of the γ1 subunit drives the relative conductance curves to more negative potentials (*Figure 2f*, left panel) compared to the values obtained without γ1 (*Figure 2f*, dashed lines). Remarkably, the change in the voltage dependence of the relative conductance induced by γ1 does not alter the simultaneously recorded FRET signals (*Figure 2f*, right panel), which remains indistinguishable from that recorded with BK667CYα (*Figure 2f*, dashed lines).

## The dynamics of the VSD are directly reflected in the gating ring conformation

Using the allosteric HA model of BK channel function, *Horrigan and Aldrich (2002)* proposed that Ca$^{2+}$ binding to the Ca$^{2+}$ bowl is coupled to the voltage sensor activation. Yet, the strength of that interaction (allosteric constant E) was smaller than those corresponding to Ca$^{2+}$- or V-sensors with channel opening (*Horrigan and Aldrich, 2002*). Interestingly, when E was derived from gating currents data, a larger value was obtained (*Carrasquel-Ursulaez et al., 2015*). Further, Ca$^{2+}$ binding to the RCK1 domain (but not to the Ca$^{2+}$ bowl) is voltage-dependent (*Sweet and Cox, 2008*), which as the authors hypothesized might originate from physical interactions between the voltage sensors and the RCK1 domains. Additionally, using the cut-open oocyte voltage-clamp fluorometry approach, *Savalli et al. (2012)* showed that fluorescence emission from reporters within the VSD could change upon uncaged Ca$^{2+}$ stimuli. This evidence indicates that the VSD is coupled to the gating ring, but none of these approaches directly monitored the conformational changes of the gating ring structure. Therefore, we decided to explore whether the voltage dependence of the gating ring movements is attributable to the voltage sensor activation. To this end we modified the voltage dependence of the VSD activation by co-expression with β auxiliary subunits or by introducing specific mutations in the VSD (*Figure 3* and *Figure 4*). The effects of co-expressing BK α subunit with the four different types of auxiliary β subunits have been extensively studied (*Tseng-Crank et al., 1996*; *Behrens et al., 2000*; *Brenner et al., 2000*; *Cox and Aldrich, 2000*; *Uebele et al., 2000*; *Lingle et al., 2001*; *Zeng et al., 2001*; *Bao and Cox, 2005*; *Orio and Latorre, 2005*; *Yang et al., 2008a*; *Sweet and Cox, 2009*; *Contreras et al., 2012*; *Li and Yan, 2016*). β1 subunit has been previously proposed to alter the voltage sensor-related voltage dependence, as well as the intrinsic opening of the gate and Ca$^{2+}$ sensitivity (*Figure 3a*) (*Cox and Aldrich, 2000*; *Bao and Cox, 2005*; *Orio and Latorre, 2005*; *Sweet and Cox, 2009*; *Contreras et al., 2012*; *Castillo et al., 2015*). Recordings from BK667CYα co-expressed with β1 subunits reveal the expected modifications in the voltage dependence of the relative conductance, that is an increase in the apparent Ca$^{2+}$ sensitivity (*Figure 3b*, left panel) (*Wallner et al., 1995*; *Cox and Aldrich, 2000*; *Bao and Cox, 2005*; *Orio and Latorre, 2005*; *Sweet and Cox, 2009*; *Contreras et al., 2012*). In addition, it has been reported that β1 subunit alters the function of the VSD (*Orio and Latorre, 2005*; *Castillo et al., 2015*). Notably, the *E*-V curves are shifted to more negative potentials (*Figure 3b*, right panel), similarly to the described modification (*Castillo et al., 2015*). The structural determinants of the β1 subunit influence on the VSD reside within its N-terminus, which has been shown by engineering a chimera between the β3b subunit (which does not influence the VSD) and the N-terminus of the β1 (β3bNβ1) (*Castillo et al., 2015*). We recapitulated this strategy. First, we co-expressed BK667CY α subunits with β3b and observed the expected inactivation of the ionic currents at positive potentials, yet with different blockade kinetics (see *Figure 3—figure supplement 1*) (*Uebele et al., 2000*; *Xia et al., 2000*; *Lingle et al., 2001*). The relative open probability of this complex is like BK667CYα alone, except that at extreme positive potentials the values of relative conductance at the tails decrease due to inactivation (*Figure 3—figure supplement 1b*, left panel). The values of *E* vs V remained comparable to those observed for BK667CYα (*Figure 3—figure supplement 1b*, right panel). We then co-expressed the β3bNβ1 chimera (*Castillo et al., 2015*) with BK667CYα (*Figure 3c*). This complex did not modify the relative conductance *vs.* voltage relationship (*Figure 3d*, left panel) as compared with BK667CYα alone (*Figure 3d*, grey shadow). On the other hand, while the magnitude of the FRET change is the same as in BK667CYα, the voltage dependence of *E* values at [Ca$^{2+}$] of 4 μM, 12 μM and 22 μM shifted to more negative potentials compared to the values of BK667CYα alone (*Figure 3d*, right panel, compare dashed to solid lines). Altogether, these results indicate that

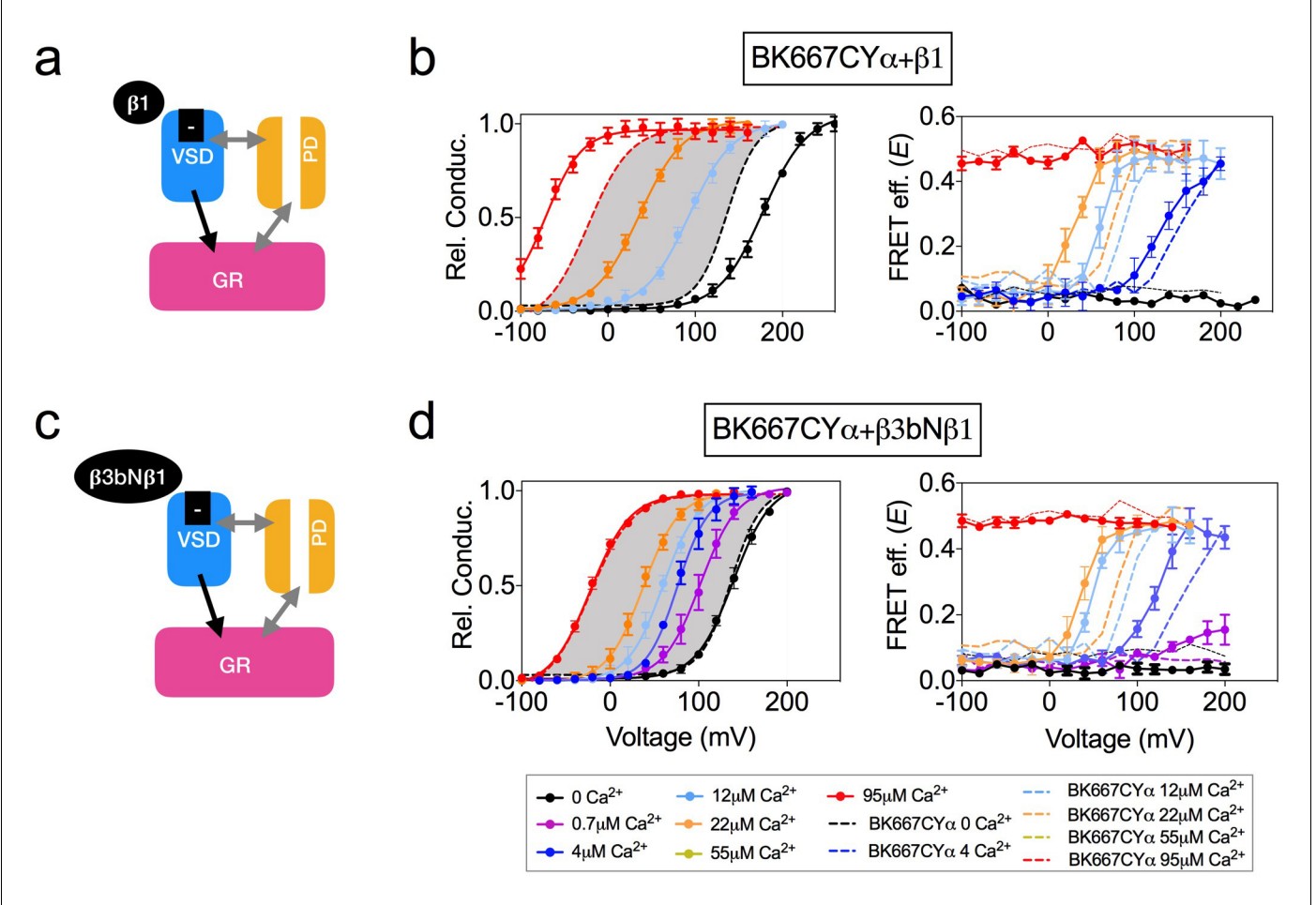

**Figure 3.** Co-expression with β subunits. (a) β1 subunits have been shown to directly regulate VSD function, shifting $V_{h(j)}$ to more negative values (b) Left panel, G-V curves obtained at several $Ca^{2+}$ concentrations after co-expression of BK667CY with the β1 subunit, which induces a leftward shift in the *E*-V curves obtained simultaneously (right). (c) β3bNβ1 chimeras produce similar effects to β1 on VSD function, since they retain the N-terminal region of β1 (*Castillo et al., 2015*). (d) G-V (left) and *E*-V curves (right) of BK667CY α subunits co-expressed with the β3bNβ1 chimera. Data corresponding to each $Ca^{2+}$ concentration are color-coded as indicated in the legend at the bottom. Colored dashed lines represent the G-V and *E*-V curves corresponding to BK667CYα channels (*Miranda et al., 2013*; *Miranda et al., 2016*). The solid curves in the G-V graphs represent Boltzmann fits. The full range of G-V curves from 0 μM $Ca^{2+}$ to 95 μM $Ca^{2+}$ from BK667CY is represented as a grey shadow in left panels (**b** and **d**), for reference. Data points and error bars represent average ± SEM ($n$ = 3–10; N = 2–4).

DOI: https://doi.org/10.7554/eLife.40664.004

The following figure supplement is available for figure 3:

**Figure supplement 1.** Co-expression with β3b subunits.

DOI: https://doi.org/10.7554/eLife.40664.005

the alteration of the voltage dependence of the voltage sensor induced by the amino terminal of β1within the β3bNβ1 chimera underlies the modification of the voltage dependence of the gating ring conformational changes, reinforcing the hypothesis that this voltage dependence is directly related to VSD function.

VSD activation can also be altered by introducing single point mutations that modify the voltage of half activation of the voltage sensor, $V_h(j)$. This parameter is determined by fitting data to the HA allosteric model (*Ma et al., 2006*) or directly from gating current measurements (*Zhang et al., 2014*). Mutations of charged amino acids on the VSD have been reported to produce different modifications in the $V_h(j)$ values. In some cases, other parameters related to BK channel activation are additionally affected by the mutations. Mutation R210E shifts the $V_h(j)$ value from +173 mV to +25 mV at 0 $Ca^{2+}$ in BK channels (*Figure 4a*) (*Ma et al., 2006*). Consistent with this, introduction of this

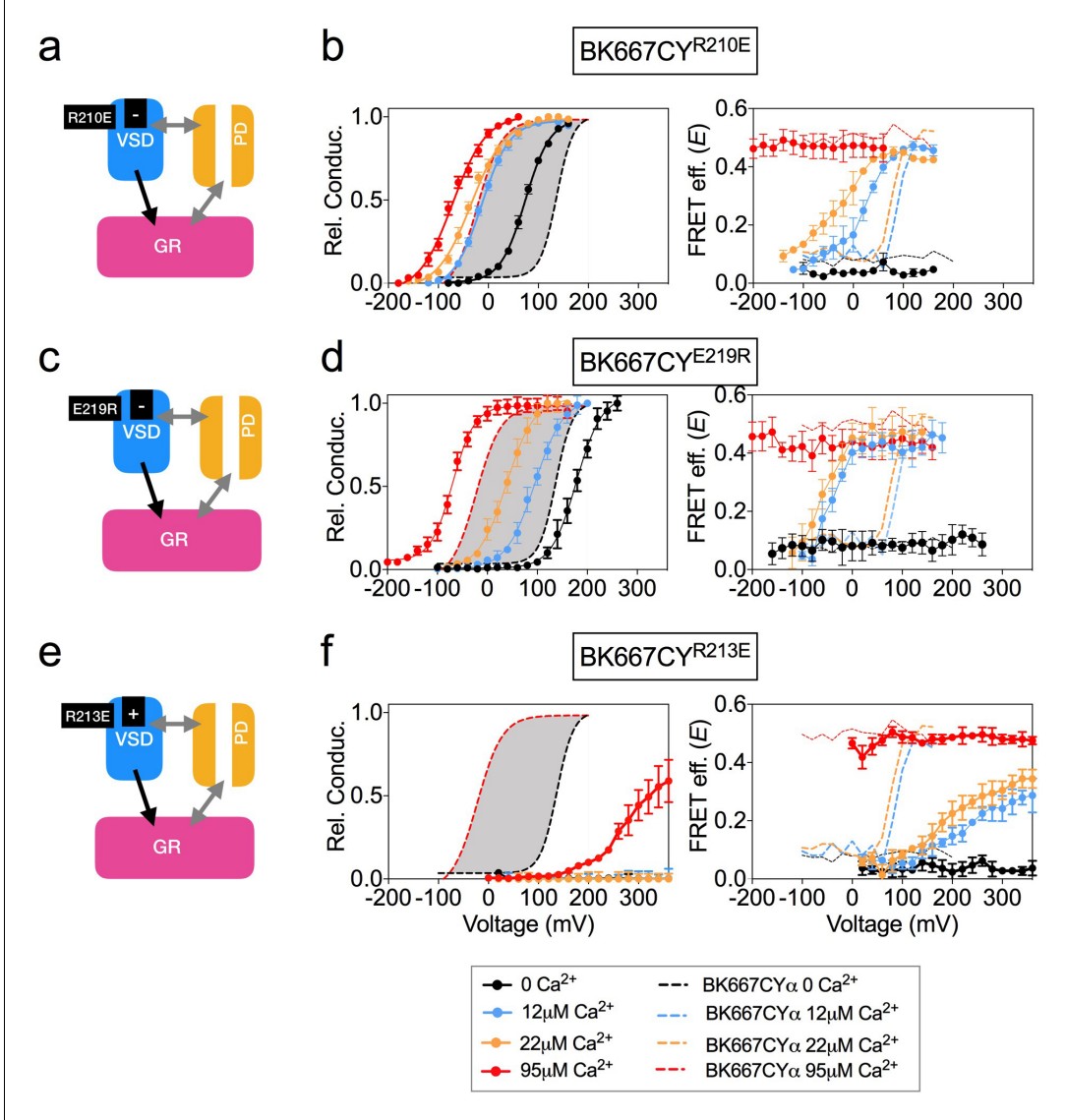

**Figure 4.** Mutation of charged residues of BK VSD. VSD activation was altered by mutation of charged residues in the VSD that modify its voltage of half activation, $V_{h(j)}$ (a) The R210E mutation induces a negative shift of $V_{h(j)}$(b) G-V (left panel) and *E*-V curves (right panel) obtained simultaneously from constructs BK667CY containing the R210E mutation at several $Ca^{2+}$concentrations. (c) The E219R mutation produces a negative shift of $V_{h(j)}$ (d) G-V (left panel) and *E*-V curves (right panel) obtained simultaneously from constructs BK667CY containing the E219R mutation at several $Ca^{2+}$concentrations. (e) The R213E mutation induces a large positive shift of $V_{h(j)}$ values. (f) G-V (left panel) and *E*-V curves (right panel) obtained simultaneously from constructs BK667CY containing the R213E mutation at several $Ca^{2+}$concentrations. Data corresponding to each $Ca^{2+}$ concentration are color-coded as indicated in the bottom legend. Colored dashed lines represent the G-V and *E*-V curves corresponding to non-mutated BK667CYα channels (*Miranda et al., 2013*; *Miranda et al., 2016*). The solid curves in the G-V graphs represent Boltzmann fits. The full range of G-V curves from 0 μM $Ca^{2+}$ to 95 μM $Ca^{2+}$ from BK667CY is represented as a grey shadow in left panels (b), (d and f), for reference. Data points and error bars represent average ± SEM ($n$ = 4–10; N = 3–4).

DOI: https://doi.org/10.7554/eLife.40664.006

mutation in BK667CYα (BK667CY$^{R210E}$) caused a shift of the relative conductance *vs.* voltage dependence towards more negative potentials (*Figure 4b*, left panel) as compared to BK667CY (*Figure 4b*, left panel, grey shadow). Simultaneously measured *E* values showed a negative shift in the voltage dependence of the FRET signal at intermediate $Ca^{2+}$ concentrations (*Figure 4b*, right panel). Mutation E219R had been previously shown to produce a large negative shift in $V_h(j)$ from +150 mV to +40 mV ($\Delta V_h(j)$ = −110 mV; *Figure 4c*), additionally modifying the $Ca^{2+}$ sensitivity

and the coupling between the VSD and channel gate (*Zhang et al., 2014*). As previously reported, BK667CY$^{E219R}$ showed modified relative conductance *vs.* voltage relationships at different $Ca^{2+}$ concentrations (*Figure 4d*, left panel) (*Zhang et al., 2014*). In addition, this construct revealed a shift to more negative potentials in the *E vs.* voltage dependence at intermediate $Ca^{2+}$ concentrations (12 μM and 22 μM $Ca^{2+}$; *Figure 4d*, right panel), paralleling the reported negative shift in $V_h(j)$ (*Ma et al., 2006*; *Zhang et al., 2014*). Since mutations displacing the $V_h(j)$ to more negative potentials induce equivalent shifts in the voltage dependence of the gating ring motion (measured as *E*), we tested if other mutations previously reported to induce positive shifts on $V_h(j)$ (*Ma et al., 2006*) were also associated with changes of the *E*-V curves in the same direction. As shown by Ma et al., the largest effect on $V_h(j)$ is induced by the R213E mutation, producing a shift of $\Delta V_h(j)=+337$ mV (*Figure 4e*) (*Ma et al., 2006*). The BK667CY$^{R213E}$ construct showed a significant shift in the voltage dependence of the relative conductance to more positive potentials (*Figure 4f*, left panel). Notably, this effect was paralleled by a large displacement in the *E vs.* voltage dependence towards more positive potentials (*Figure 4f*, right panel). Taken together, our data show that modifications of the $V_h(j)$ values caused by mutating the VSD charged residues are reflected in equivalent changes in the voltage dependence of the gating ring conformational rearrangements, which occur in analogous directions and with proportional magnitudes at intermediate $Ca^{2+}$ concentrations.

All these results on the VSD modifications and their corresponding changes in FRET signals support the existence of a direct coupling mechanism between the VSD function and the gating ring conformational changes.

### Parallel alterations of the voltage dependence of VSD function and gating ring motions by selective activation of the RCK1 binding site

We have previously shown that specific interaction of $Cd^{2+}$ with the RCK1 binding site leads to activation of the BK channel, which is accompanied by voltage-dependent changes in the *E* values at intermediate $Cd^{2+}$ concentrations of 10 μM and 30 μM (*Miranda et al., 2016*). To further assess the role of the RCK1 binding site activation in the voltage dependence of the gating ring motions, we studied activation by $Cd^{2+}$ of selected BK667CY VSD mutants (*Figure 5*). Addition of $Cd^{2+}$ to the BK667CY$^{E219R}$ mutant (*Figure 5a*) shifted the voltage dependence of *E* towards more negative potentials at intermediate $Cd^{2+}$ concentrations (10 μM and 30 μM; *Figure 5b*) when compared to non-mutated BK667CY (*Figure 5b*; dashed lines). This change in the *E*-V curves induced by selective activation of the RCK1 binding site with $Cd^{2+}$ paralleled the large negative shift ($\Delta V_h(j) = -110$ mV) previously reported with the E219R mutant BK channels (*Ma et al., 2006*; *Zhang et al., 2014*). We also tested $Cd^{2+}$ activation in the mutant BK667CY$^{R201Q}$, which shifts the $V_h(j)$ parameter by 47 mV towards positive potentials (*Figure 5c*) (*Ma et al., 2006*). Addition of $Cd^{2+}$ rendered right-shifted *E* vs. voltage relationships (*Figure 5d*, right panel), following the direction of the predicted $V_h(j)$ shift described for this mutant BK channel (*Ma et al., 2006*). Finally, addition of $Cd^{2+}$ to the BK667CY$^{F315A}$ construct (*Figure 5e*) (*Carrasquel-Ursulaez et al., 2015*) did not have any effect on the *E*-V relationship (*Figure 5f*). These results are consistent with a mechanism in which specific binding of $Cd^{2+}$ to the RCK1 binding site allows voltage-dependent conformational changes in the gating ring that are directly related to VSD activation.

### Voltage dependence of $Ba^{2+}$-induced gating ring movement is related to function of the channel gate

$Ca^{2+}$, $Mg^{2+}$ and $Ba^{2+}$ bind to the $Ca^{2+}$ bowl and trigger conformational changes of the gating ring region (*Miranda et al., 2016*). However, the effects of these ions on BK function and gating ring motions are fundamentally different. Notably, $Ba^{2+}$ induces a rapid blockade of the BK current after a transient activation that is measurable at low $Ba^{2+}$ concentrations (*Zhou et al., 2012*; *Miranda et al., 2016*) (*Figure 6a*). In addition, we previously showed that the gating ring conformational motions induced by $Ba^{2+}$ show a voltage-dependent component, which is not observed when $Ca^{2+}$ or $Mg^{2+}$ bind to the $Ca^{2+}$ bowl (*Miranda et al., 2013*; *Miranda et al., 2016*) (*Figure 6b*). We combined mutagenesis with the cation-specific activation strategy to identify the structural source of the voltage dependence in $Ba^{2+}$-triggered gating ring motions. In this case, alteration of VSD function by mutating charged residues (*Figure 6c and e*) was not reflected in any change of the *E vs.* voltage relationships, as shown in *Figure 6d and f* for constructs BK667CY$^{R210E}$ and BK667CY$^{R213E}$,

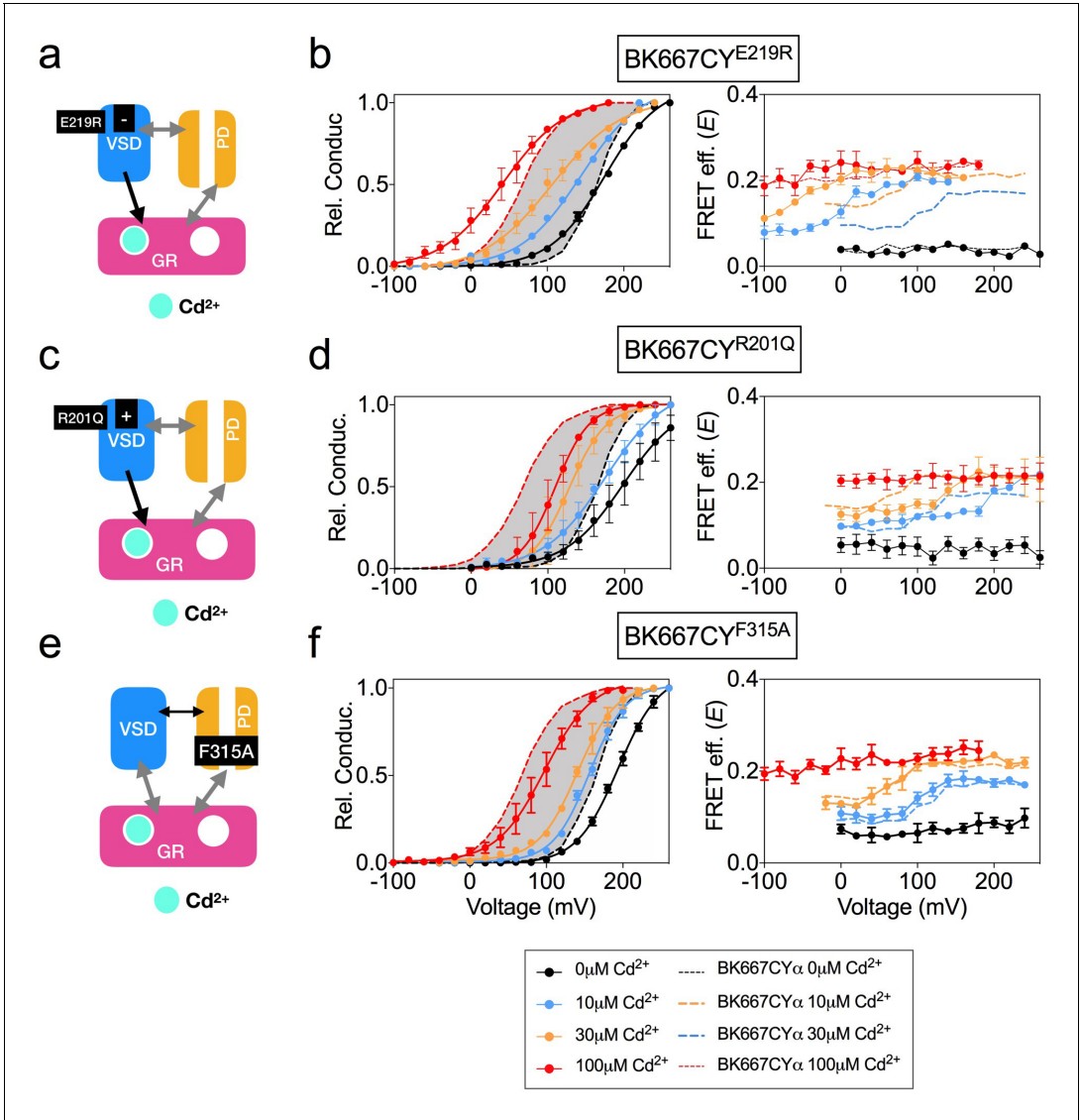

**Figure 5.** Voltage dependence of gating ring rearrangements after specific activation of RCK1 high-affinity binding site by $Cd^{2+}$. (a) Effect of the VSD E219R mutation on the selective activation of RCK1 by $Cd^{2+}$. (b) G-V (left panels) and *E*-V curves (right panels) obtained simultaneously at several $Ca^{2+}$concentrations from constructs BK667CY[E219R]. (c) VSD R201Q mutation induces a positive shift of $V_{h(j)}$ (d) G-V (left panels) and *E*-V curves (right panels) obtained simultaneously at several $Cd^{2+}$concentrations from constructs BK667CY[R201Q] (e) Effect of the F315A mutation on the selective activation of RCK1 by $Cd^{2+}$. (f) G-V (left panels) and *E*-V curves (right panels) obtained simultaneously at several $Cd^{2+}$ concentrations from constructs BK667CY[F315A]. Data corresponding to each $Cd^{2+}$ concentration are color-coded as indicated in the legend at the bottom. Colored dashed lines represent the G-V and *E*-V curves corresponding to BK667CYα channels (*Miranda et al., 2013*; *Miranda et al., 2016*). The solid curves in the G-V graphs represent Boltzmann fits. The full range of G-V curves from 0 μM $Cd^{2+}$ to 100 μM $Cd^{2+}$ corresponding to non-mutated BK667CY is represented as a grey shadow in left panels (b), (d), and (f), for reference. Data points and error bars represent average ± SEM (*n* = 3–4; N = 2).
DOI: https://doi.org/10.7554/eLife.40664.007

respectively. These results indicate that the voltage dependence of $Ba^{2+}$-induced gating ring conformational changes, unlike those induced by $Ca^{2+}$ and $Cd^{2+}$ through activation of the RCK1 binding site, may not be related to VSD activation. This conclusion is further supported by the lack of changes in $Ba^{2+}$ responses when mutations in the VSD were made in a RCK1 $Ca^{2+}$ binding site knockout (D362A D367A) background (*Figure 6—figure supplement 1b & c*). Next, we studied the effect of $Ba^{2+}$ on BK667CY channels containing the F315A mutation (*Figure 6g*) (*Carrasquel-Ursulaez et al., 2015*). As shown in *Figure 6h*, the *E* values reached similar levels to those of non-mutated BK667CY channels at saturating $Ba^{2+}$ concentrations. However, at intermediate

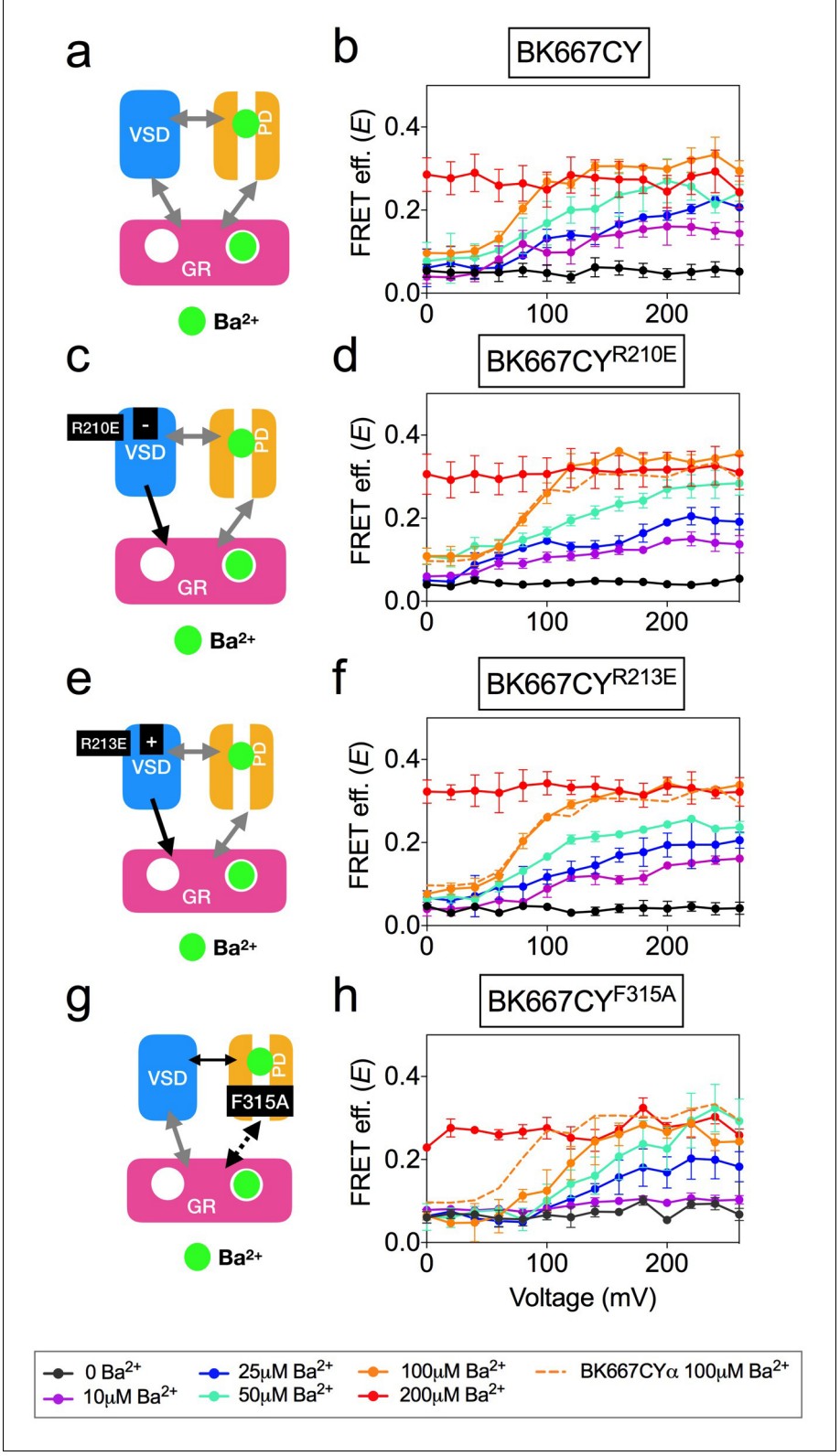

**Figure 6.** Voltage dependence of gating ring movements triggered by $Ba^{2+}$. (**a**) The RCK2 site is selectively activated by $Ba^{2+}$, which additionally induces pore block. (**b**) FRET efficiency ($E$) data obtained at several $Ba^{2+}$ concentrations from BK667CY constructs (*Miranda et al., 2016*). (**c**) Effect of the VSD R210E mutation after selective activation of the RCK2 binding site by $Ba^{2+}$. (**d**) $E$-V curves obtained at several $Ba^{2+}$ concentrations from *Figure 6 continued on next page*

*Figure 6 continued*

BK667CY$^{R210E}$ constructs. (**e**) Effect of the VSD R213E mutation after selective activation of the RCK2 binding site by Ba$^{2+}$. (**f**) *E*-V curves obtained at several Ba$^{2+}$ concentrations from BK667CY$^{R213E}$ constructs. (**g**) Effect of the F315A mutation after selective activation of the RCK2 binding site by Ba$^{2+}$ (**h**) *E*-V curves obtained at several Ba$^{2+}$ concentrations from BK667CY$^{F315A}$ constructs. Data corresponding to each Ba$^{2+}$ concentration are color-coded according to the legend at the bottom. For reference, the curve corresponding to 100 µM Ba$^{2+}$ from the BK667CY construct shown in (**b**) is also shown as a colored dashed line in panels (**b, d, f** and **h**). Data points and error bars represent average ± SEM (*n* = 4–6; N = 2–3).

DOI: https://doi.org/10.7554/eLife.40664.008

The following figure supplement is available for figure 6:

**Figure supplement 1.** Additional experiments to characterize voltage dependence of gating ring movements triggered by Ba$^{2+}$.

DOI: https://doi.org/10.7554/eLife.40664.009

---

concentrations of Ba$^{2+}$ the *E*-V curves were shifted towards more positive potentials when compared with BK667CY channels (*Figure 6h*, dashed line). These results suggest that the voltage-dependent component of the conformational changes triggered by Ba$^{2+}$ binding to the Ca$^{2+}$ bowl are not directly related to VSD activation, but rather to the function of the channel gate.

## Discussion

Using fluorescently labeled BKα subunit constructs reporting protein dynamics between the RCK1 and RCK2 domains, we previously demonstrated that the channel high-affinity binding sites can be independently activated by different divalent ions, inducing energetically-additive rearrangements of the gating ring measured as changes in the FRET efficiency values, *E* (*Miranda et al., 2013*; *Miranda et al., 2016*). Further, the effects of Ca$^{2+}$, Cd$^{2+}$ and Ba$^{2+}$ on the *E* values showed a voltage-dependent component, for which we could not provide an explanation. Voltage dependence of Ca$^{2+}$-induced rearrangements seemed to be specifically related to RCK1 activation, since only the mutation of that binding site resulted in voltage-independent *E* signals (*Miranda et al., 2016* and *Figure 1*). One possibility to explain this result is the existence of direct structural interactions of the RCK1 domain and the VSD. Interestingly, the recently obtained cryo-EM full BK structure from *Aplysia californica* revealed the existence of specific protein-protein interfaces formed by the amino terminal lobes of the RCK1 domains facing the transmembrane domain and the VSD/S4-S5 linkers (*Hite et al., 2017*). According to the structural data obtained in saturating Mg$^{2+}$ and Ca$^{2+}$ concentrations, gating of the channel by Ca$^{2+}$ was proposed to be mediated, at least partly, by displacement of these interfaces causing the VSD/S4-S5 linkers to move, contributing to pore opening ((*Hite et al., 2017*; *Tao et al., 2017*); but see also (*Zhou et al., 2017*)). Our work provides functional data supporting this mechanism. Our data show that mutations altering the voltage dependence of BK VSD are reflected in the voltage dependence of the gating ring movements triggered by activation of the RCK1 binding site by Ca$^{2+}$ or Cd$^{2+}$. Mutations altering VSD function by inducing large leftward shifts in the V$_h$(j) values (*Ma et al., 2006*; *Zhang et al., 2014*) strongly correlate with negative shifts in the voltage dependence of the *E* signals. Likewise, mutations inducing positive shifts in the VSD voltage dependence of the voltage sensor function are reflected in *E*-V shifts towards more positive membrane voltages. Interestingly, we also observe a correlation between the changes in the slope of the G-V curves and that of the *E*-V curves (e.g. *Figure 4f*; see also *Supplementary file 1*), suggesting the existence of an interaction between the VSD and the gating ring. This idea is further supported by the effect of β1 which has been proposed to alter the voltage dependence of VSD function (*Wallner et al., 1995*; *Cox and Aldrich, 2000*; *Nimigean and Magleby, 2000*; *Bao and Cox, 2005*; *Orio and Latorre, 2005*; *Contreras et al., 2012*; *Castillo et al., 2015*). We observed that β1 and β3bNβ1 induce a leftward shift in the *E*-V curves. Conversely, two experimental strategies known to influence the G-V curves without direct interference with the VSD did not affect the voltage dependence of *E*. The lack of effect on the *E*-V curves of the mutation F315A can be explained because the shift in the G-V curves arises from the influence of this mutation in the C⟷O transition with minor effects on the voltage dependence of the gating currents (*Carrasquel-Ursulaez et al., 2015*). Analogously, no change in the voltage dependence of *E* was observed after

co-expression of BKα with the γ1 subunit, which shifts the voltage dependence of pore opening by enhancing its allosteric coupling with the voltage sensor activation (*Yan and Aldrich, 2010*). As with the mutation F315A, the presence of γ1 subunit produces a minor shift in the Q-V distributions, not paralleling the large shift in the G-V curves (Carrasquel-Ursulaez and Ramon Latorre, personal communication).

A puzzling result from our previous study was the observation that $Ba^{2+}$ binding to the $Ca^{2+}$ bowl triggers voltage-dependent conformational changes (*Miranda et al., 2016*). Even though we still do not know the mechanisms of this unique response to $Ba^{2+}$, here we learned that it is not related to the dynamics of VSD, but rather influenced by perturbations affecting the opening and closing of the channel at the pore domain. Why $Ba^{2+}$ but not $Ca^{2+}$? A possible answer for this question is that $Ba^{2+}$ has the additional property of blocking the permeation pathway (*Miller, 1987*; *Neyton and Miller, 1988*; *Zhou et al., 2012*), which could somehow be transmitted allosterically to the gating ring. If simply ion permeation blockade is what matters, then we might expect that blocking permeation with the high affinity quaternary ammonium derivative *N*-(4-[benzoyl]benzyl)-*N,N,N*-tributylammonium (bb-TBA) (*Tang et al., 2009*) should produce a voltage dependent FRET signal with $Ca^{2+}$ activation. But, it does not (*Figure 6—figure supplement 1d*). Another possibility for the $Ba^{2+}$ effect could be a direct allosteric interaction between the intrinsic gating in the pore and the divalent binding site in RCK2, which needs to be tested further.

Irrespectively of the fluorescent construct (*Miranda et al., 2013*) or the divalent ion used to activate the BK channel (*Miranda et al., 2016*), we have consistently observed that the conformational changes monitored as changes in the FRET efficiency are not strictly coupled to the intrinsic gating of the channel. In this study, we have found that the consequences of the voltage dependence of the intrinsic gating by manipulations of the VSD and the pore region are paralleled by the FRET efficiencies. These results rule out the possibilities that FRET signals derive from conformational changes in an unknown $Ca^{2+}$ binding site or that they are completely uncoupled to the intrinsic gating.

In conclusion, our functional data show a strong correlation between the VSD function and the RCK1 conformational changes, suggesting a transduction mechanism from ion binding to change the channel activation. This transduction mechanism is in agreement with the existence of structural interactions between the RCK1 domain and the VSD. The correlation between VSD function and the RCK1 conformational changes is not observed between RCK2 and VSD, suggesting the existence of a different transduction mechanism that may include an indirect mechanism through the RCK1 or RCK1-S6 linker.

# Materials and methods

## Molecular biology and heterologous expression of tagged channels

Fluorescent BK α subunits were labelled with CFP or YFP using a transposon-based insertion method (*Giraldez et al., 2005*). Subunits labelled in the position 667 were subcloned into the pGEMHE oocyte expression vector (*Liman et al., 1992*). RNA was transcribed in vitro with T7 polymerase (Ambion, Thermo Fisher Scientific, Waltham, USA), and injected at a ratio 3:1 of CFP: YFP into *Xenopus laevis* oocytes, giving a population enriched in 3CFP:1YFP labelled tetramers (BK667CY) (*Miranda et al., 2013*; *Miranda et al., 2016*). Individualized Oocytes were obtained from *Xenopus laevis* extracted ovaries (Nasco, Fort Anderson, WI, USA). Neutralization of the $Ca^{2+}$ bowl was achieved by mutating five consecutive aspartate residues to alanines (5D5A: 894–899) (*Bao et al., 2002*) on the BK667CY background. Elimination of RCK1 high-affinity $Ca^{2+}$ sensitivity was achieved by double mutation D362A and D367A (*Xia et al., 2002*; *Zeng et al., 2005*; *Zhang et al., 2010*). Mutations were performed using standard procedures (Quickchange, Agilent Technologies, Santa Clara, USA). Auxiliary subunits (β3b, γ1 and chimera β3bNβ1) were co-injected with the BK667CFP/BK667YFP RNA mix at a 5:1 wt ratio, giving molar ratios above 20:1.

## Patch-clamp fluorometry and FRET

Borosilicate pipettes with a large tip (0.7–1 MΩ in symmetrical $K^+$) were used to obtain inside-out patches excised from *Xenopus laevis* oocytes expressing BK667CY. Currents were recorded with the Axopatch 200B amplifier and Clampex software (Axon Instruments, Molecular Devices, Sunnyvale,

USA). Recording solutions contained (in mM): pipette, 40 KMeSO$_3$, 100 N-methylglucamine-MeSO$_3$, 20 HEPES, 2 KCl, 2 MgCl$_2$, 100 μM CaCl$_2$ (pH 7.4); bath solution, 40 KMeSO$_3$, 100 N-methylgluc-amine-MeSO$_3$, 20 HEPES, 2 KCl, 1 EGTA, and MgCl$_2$ or BaCl$_2$ to give the appropriate divalent concentration previously estimated using Maxchelator software (maxchelator.standford.edu) (*Bers et al., 1994*). Solutions containing Cd$^{2+}$ were prepared with a bath solution containing KF instead of K-Mes to precipitate the contaminant Ca$^{2+}$ previously to the administration of the proper concentration of CdCl$_2$ estimated with Maxchelator. Solutions containing different ion concentrations were exchanged using a fast solution-exchange system (BioLogic, Claix, France). All experiments were performed in various batches of oocytes, using different Ca$^{2+}$ solutions prepared over time.

Simultaneous fluorescent and electrophysiological recordings were obtained as previously described (*Miranda et al., 2013*; *Miranda et al., 2016*). Conductance-voltage (G-V) curves were obtained from tail currents using standard procedures. The G-V relations were fit with the Boltzmann function: G/Gmax = 1/(1 + exp (-zF(V-Vhalf)/RT), where Gmax is the maximum tail current, z is the voltage dependence of activation, V$_{half}$ is the half-activation voltage of the ionic current. T is the absolute temperature (295K), F is the Faraday's constant and R the universal gas constant. Fit parameters are provided in *Supplementary file 1*. Conformational changes of the gating ring were tracked as intersubunit changes of the FRET efficiency between CFP and YFP as previously reported (*Miranda et al., 2013*; *Miranda et al., 2016*). Analysis of the FRET signal was performed using emission spectra ratios. We calculated the FRET efficiency as $E$=(RatioA-RatioA$_0$)/(RatioA$_1$-RatioA$_0$), where RatioA and RatioA$_0$ are the emission spectra ratios for the FRET signal and the control only in the presence of acceptor respectively (*Zheng and Zagotta, 2003*); RatioA$_1$ is the maximum emission ratio that we can measure in our system (*Miranda et al., 2013*; *Miranda et al., 2016*). This value of $E$ is proportional to FRET efficiency (*Zheng and Zagotta, 2003*). The $E$ value showed is an average of the $E$ value corresponding to each tetramer present in the membrane patch and represent an estimation of the distance between the fluorophores located in the same position of the four subunits of the tetramer. Where possible, the $E$-V relations were fit with the Boltzmann function: $E$ = 1/(1 + exp (-zF(V-Vhalf)/RT), where z is the voltage dependence of the gating ring movement ($E$) and V$_{half}$ is the half-activation voltage of the fluorescent signal. Fit parameters are provided in *Supplementary file 1*.

## Acknowledgments

MH and PM were supported by the intramural section of the National Institutes of Health (NINDS). TG was funded by the Spanish Ministry of Economy and Competitivity (grants SAF2013-50085-EXP and RyC-2012–11349) and the European Research Council (ERC) under the European Union's Horizon 2020 research and innovation programme (grant agreement 648936). We thank Deepa Srikumar for technical assistance and Andrew Plested for useful comments on the manuscript. The γ1 clone and the β3bNβ1 chimera were kind gifts from Chris Lingle and Ramon Latorre, respectively.

## Additional information

### Funding

| Funder | Grant reference number | Author |
| --- | --- | --- |
| National Institute of Neurological Disorders and Stroke | ZIA-NS002993 | Pablo Miranda Miguel Holmgren |
| H2020 European Research Council | ERC-CoG-2014-648936 | Teresa Giraldez |
| Ministerio de Economía y Competitividad | SAF2013-50085-EXP | Teresa Giraldez |
| Ministerio de Economía y Competitividad | RyC-2012-11349 | Teresa Giraldez |

The funders had no role in study design, data collection and interpretation, or the decision to submit the work for publication.

## Author contributions
Pablo Miranda, Conceptualization, Resources, Data curation, Formal analysis, Investigation, Visualization, Writing—original draft, Project administration, Writing—review and editing; Miguel Holmgren, Conceptualization, Resources, Formal analysis, Supervision, Funding acquisition, Visualization, Writing—original draft, Project administration, Writing—review and editing; Teresa Giraldez, Conceptualization, Formal analysis, Supervision, Funding acquisition, Visualization, Methodology, Writing—original draft, Project administration, Writing—review and editing

## Author ORCIDs
Teresa Giraldez http://orcid.org/0000-0002-4096-810X

## Decision letter and Author response
Decision letter https://doi.org/10.7554/eLife.40664.013
Author response https://doi.org/10.7554/eLife.40664.014

## Additional files

### Supplementary files
• Supplementary file 1. Fit parameters of data shown in *Figures 1–6*. The G-V and *E*-V relations were fit with Boltzmann functions G/Gmax = 1/(1 + exp (-zF(V-Vhalf)/RT), *E* = 1/(1 + exp (-zF(V-Vhalf)/RT), where Gmax is the maximum tail current, z is the voltage dependence of activation (G) or gating ring movement (*E*), Vhalf is the half-activation voltage of the ionic current or the fluorescent signal. T is the absolute temperature (295K), F is the Faraday's constant and R the universal gas constant.
DOI: https://doi.org/10.7554/eLife.40664.010
• Transparent reporting form
DOI: https://doi.org/10.7554/eLife.40664.011

### Data availability
All data generated and analysed during this study are included in the manuscript.

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
