## [Decision Letter]

Thank you for submitting your article "The BK channel gating ring is strongly coupled to the voltage sensor" for consideration by *eLife*. Your article has been reviewed by three peer reviewers, and the evaluation has been overseen by Richard Aldrich as the Senior and Reviewing Editor. The following individual involved in review of your submission has agreed to reveal his identity: Christopher J Lingle (Reviewer #2).

The reviewers have discussed the reviews with one another and the Reviewing Editor has drafted this decision to help you prepare a revised submission.

Essential revisions:

1) Modify the Title

Reviewers thought the title was vague and could be modified to better convey the message of the manuscript. The term "strongly coupled" also needs to be modified or justified since it implies a quantitative assessment of coupling that was not addressed.

2) Address previous work and novelty

The authors should provide a more complete comparison with previous work, especially with regards to VSD/gating-ring coupling or interaction, to help clarify "what is new" and whether the results are consistent or inconsistent with previous results.

3) Provide more guidance for interpreting the results

Indicating the expected status of voltage sensors under different conditions and manipulations would help readers to interpret the corresponding FRET data. Reviewers also desired more guidance on how to interpret the quantitative relationship between VSD activation and the voltage-dependence of FRET as well as several interesting or puzzling features of the data including the effect of Ba^2+^, the remarkably steep Ca^2+^ dependence of FRET, and the lack of voltage-dependence of FRET in 0 Ca^2+^. Mechanistic or quantitative interpretation is lacking, and ideally should be improved, if only to provide some framework for future evaluation and tests. Proposing a model (i.e. a kinetic scheme) to account for the major features of the relation between Ca^2+^, voltage, and FRET efficiency (i.e. lack of FRET in 0 Ca^2+^ or high Ca^2+^, and correlations observed in the different mutants) would greatly strengthen the conclusions and enhance the significance of the results. While developing a model to account for all features of the data may be beyond the scope of this study, the authors should at minimum indicate which features of the data can or cannot be accounted for by previously proposed models (i.e. Miranda 2013) and whether their data provide any new insight into mechanism that must be included in gating models. This should be possible to do without incorporating additional experiments.

4) Reconcile discrepancies and apparent inconsistencies in the data

The authors should address the apparent inconsistency between the effect of the γ1 subunit on FRET and its predicted effect on VSD activation, as well as discrepancies between some of the results and data previously published.

Results which seem to contradict earlier results include: a) GV shift magnitude of F315A in 100 Ca^2+^; b) Lack of a GV shift for the β3Nβ1 construct; c) Whether they have adequate expression of β3b.

Because the individual reviews include several important points they are included here for you reference.

Reviewer #1:

The manuscript by Miranda et al. examines energetic coupling in BK channels between its modular domains that underlie voltage sensing and calcium sensing, using patch-clamp fluorometry. The authors show here that the FRET efficiency (E) depends strongly on the integrity of the Ca^2+^ binding site in RCK1 (determined by D362/D367), and relatively much less on integrity of the Ca^2+^ bowl site. It is further shown that movements in the gating ring that are tracked by the authors' FRET measurements are not directly linked to channel opening, but instead to voltage-sensor dynamics. The greater relative contribution of the RCK1 site to VSD coupling is corroborated by selective activation of the site by Cd^2+^. However (interestingly), selective activation of the Ca^2+^ bowl site by Ba^2+^ also exhibits voltage-dependence in the FRET response, which is attributed energetic coupling with the movement of the gate. These results thus substantially extend previous work on preferential coupling of Ca^2+^ binding at the RCK1 site with voltage-sensor activation.

Overall, the manuscript is written in a logical format and the data, as presented, are of high quality. This is an outstanding contribution to our understanding of BK channels, forming a bridge between structural and functional studies. I have only a few comments aimed at improving the presentation.

1) The title seems a little vague; maybe the authors could think of one that more strongly conveys the message of this paper, which appears to be that the coupling is preferential, or selective, among the Ca^2+^-activation sites.

2) Unfortunately the authors measurement of FRET efficiency is represented by "E", and the allosteric coupling between the gating ring and voltage sensor from the Horrigan model is also "E". This makes things a bit confusing, but I'm not sure what can be done about it.

3) In Discussion (paragraph 1):

• "inducing additive rearrangements" should probably read "inducing energetically-additive rearrangements".

• " Further, the effects of Ca^2+^, Cd^2+^ and Ba^2+^ on the *E* values showed a voltage-dependent component, for which we could not provide a structural basis." This statement seems a little misleading; there seems to be some structural basis for voltage-dependence of Ca^2+^ and Cd^2+^ effects, because the RCK1 sight is physically somewhat close to the VSD (as the authors reason later on), and it is the Ba^2+^ effect that is more puzzling.

4) It seems like the interpretation of the Ba^2+^ effect may require more study, or more comment; if Ba^2+^ really is acting solely through the Ca^2+^ bowl, then the E-V relation observed with Ba and the S4 mutants (Figure 6D and F) should resemble the E-V relations with an RCK1 site knockout (D362A/D367A) combined with S4 mutation. Is this the case? If not, then this might argue against Ba^2^+ acting solely through the Ca^2+^ bowl, and suggest the Ba^2+^ is having other effects.

Reviewer #2:

This team has for several years been probing conformational changes in the cytosolic gating ring of the BK channel using sets of genetic encoded fluorescent tags to generate FRET signals that change with conformational status. Previous results had revealed an unexplained result that the FRET-reported signals exhibited voltage-dependence, despite the fact that the gating ring is not within the electric field across the membrane. This tended to contradict a standard model of BK activation in which coupling between voltage-sensors (VSDs) and the cytosolic domain (CTD) was thought to be rather weak. Furthermore, previous results from this team indicated that the two high affinity Ca^2+^ binding sites in the gating ring may behave differently in regards to voltage-sensitivity. The present paper nicely extends these earlier results with a number of functional tests that lead to the conclusion that, in the presence of Ca^2+^, the status of the voltage-sensors (VSDs) can influence the conformational status of the gating ring. Furthermore, this largely involves Ca^2+^ ligation at the RCK1 domain, but not the Ca^2+^ bowl. In sum, the authors find that the previously reported voltage-dependent conformational change of the BK gating ring induced by Ca^2+^-binding to the RCK1 high-affinity Ca^2+^ binding site is strongly correlated with the activation of the BK VSD. Manipulations that shift the activation-voltage relationship of the BK VSD, including mutations of key residues in the BK VSD or co-expression with β-1 accessory subunits, also shift the fluorescence-voltage relationship of the BK gating ring toward same direction. Manipulations that change other aspect of BK channel gating, such as F315A mutation that reduces the intrinsic BK gating equilibrium constant, the coupling between the BK VSD and BK gate, and the coupling the BK VSD and BK gating ring, or co-expression with the γ-1 subunit that enhances the coupling between the BK VSD and BK gate, do not significantly change the fluorescence-voltage relationship of the BK gating ring.

Overall, the paper is interesting, addresses an important, unresolved question, and is clearly presented (albeit with some suggestions below). There are some details on particular experiments which may be somewhat inconsistent with some earlier results and may require clarification. This particularly applies to some of the β subunit tests.

Guidance regarding status of voltage sensors under different conditions. Some readers might also wonder about what the voltage-sensor status is under various manipulations for which GVs and EVs are generated. Although explicit measurements of gating currents would be too much to ask, there are situations where it might be useful for the readers to be reminded whether VSDs are likely to be entirely resting or mostly activated, for example. The authors are essentially assuming that the E/V measurements are a surrogate for VSD activation, but unfortunately this is largely an untested assumption, although not unreasonable. For the F315A construct, the earlier Carrasquel, 2015 paper does provide comparable Q/V curves for human F380A, which support the view that the VSD equilibrium is not altered by this mutation. However, the Vh for activation of the human F380A construct in that paper at 100 μM Ca^2+^ is about 232 mV, much more positive than what is reported (less than +100 mV) in the present manuscript for mouse F315A. Reasons for this rather major discrepancy are not clear and not mentioned.

Are CTD and VSD totally uncoupled in absence of ligand? There is one aspect of the data that seems to pose an interesting limiting condition regarding the overall mechanism that the authors might consider discussing. Specifically, in the absence of Ca^2+^, any conformational change involving VSDs and pore-gate-domain (PGD) activation seems entirely uncoupled from the gating ring. It seems impressive that for every construct that is examined, at 0 Ca^2+^, even when full activation is shifted to potentials negative to +100 mV (e.g., with γ1), there is never any detectable FRET signal. This seems to say that Ca^2+^ is required for conformational coupling between VSD and CTD, or, to say it another way, that VSD movements do not couple to the CTD in the absence of Ca^2+^, or Ca^2+^ is obligatory for VSD-CTD coupling.

Is there a way to compare Ca^2+^-dependence of EV measurements to other approaches? Another interesting aspect of the data about which little is said is the rather astonishingly steep Ca^2+^-dependence of FRET signal generation at any single voltage. This seems consistent across constructs, and is reduced in the single site constructs with Cd^2+^ and Ba^2+^. Maybe this fits with what we know about Ca^2+^-dependence of activation, but it seems unusually steep. Although it may be difficult to extract from the E./V data, it seems like there is some information regarding apparent voltage-dependence of ligand affinity embedded in the data, that might be interesting to contrast to earlier results, e.g., from Sweet and Cox.

Various aspects of the β subunit experiments are confusing. First, in Figure 3D, the β3bNβ1 construct is producing no effect on GVs at all. Yet, in the Supplementary Information of Castillo et al., 2015, Vh is reported to shift negatively about 50 mV for presumably the same construct. Part of their argument was that the positive residues in the β1 N-terminus are responsible not only for the effect on voltage-sensors, but also the gating shift. Mutation of the K3K4 charged residues in the N-terminus abolishes the β1 gating shift, while mutation of K10R11 retained the gating shift.

Figure 3—figure supplement 1. Presumably the β3b construct is the human form, but this should be clarified, since the mouse form does not cause rapid inactivation. The results in Figure 3—figure supplement 1 are only partially supportive of the idea that adequate expression of β3b has been achieved. The main argument the authors make is apparently the roll-over in the GV relationship seen at 12 and 22 microM Ca^2+^. But it is very odd that this roll-over is not seen at 95 and 4 microM. The Materials and methods state that all GVs were generated from tail currents. Because the β3b inactivation is very low affinity, it has been shown that recovery from block is essentially complete at the peak of the usual tail currents, so tail current GVs are not particularly affected by this block (Xia et al., 2000. Figure 6C vs. 6D.). Yet steady-state conductance with adequate β3b expression shows marked block, which is really not so readily apparent in the example traces in panel a. The β3 subunit, of all β subunits, is known to be a bit challenging to express and the results here are not convincing in that regards. This is critical, when the observed results are really not any different from BK667CY α alone. The fact that no gating shift was observed with the chimeric β3bNβ1 construct also raises some concern in that regard. Positive controls that these β subunit constructs are being adequately expressed are lacking.

The scale bars in panel A of Figure 3—figure supplement 1 are also likely to be in error. And the time base of these recordings really doesn't provide any chance of seeing the distinctive features of patches with β3b containing channels.

Resolution of the issues raised above would impact on the discussion about these experiments presented in the subsection “The dynamics of the VSD are directly reflected in the gating ring conformation”.

Reviewer #3:

This study examines the interaction between transmembrane and cytoplasmic domains of BK channels by determining the effects of mutations in each domain, or co-assembly with regulatory subunits, on conformational changes in the cytoplasmic gating ring. The authors have previously demonstrated that gating ring conformation can be monitored by measuring FRET efficiency between CFP- and YFP-tagged Slo1 subunits. FRET efficiency is increased by the binding of Ca^2+^ and other divalent cations to two previously identified high-affinity binding sites, and at intermediate Ca^2+^ concentrations by membrane depolarization. The effect of voltage on FRET is consistent with the suggestion, most recently based on BK channel structure, that functional interactions occur between the RCK1 domain of the gating ring and VSD. Here, this hypothesis is supported by showing that (a) voltage-dependent FRET is abolished by mutating the RCK1 Ca^2+^ binding site and (b) changes in voltage-dependent FRET induced by mutations or regulatory subunits are correlated with VSD activation rather than channel opening.

The manuscript is well written and the data clearly described. However, the conclusions are not particularly novel and I have concerns about the lack of comparison to previous work, and the extent to which the results provide new insight.

The idea that functional interactions occur between the VSD and gating ring in BK channels has already been addressed by several previous studies such as Horrigan and Aldrich, 2002, Sweet and Cox, 2008, and Savalli et al., 2012. Only the first of these is referenced in the manuscript, and that one is mischaracterized. The authors seem to suggest that an interaction between VSD and Ca^2+^-binding sites was included in the Horrigan Aldrich model simply for completeness (i.e. because it couldn't be ruled out). In fact, that study observed clear evidence for an interaction based on the effects of Ca^2+^ on VSD activation (gating current). These results led to an estimate of the interaction energy between Ca^2+^- and V-sensors and the conclusion that the interaction was not altered by channel opening. In addition, Horrigan and Aldrich concluded that the interaction was weak, not because they couldn't measure it, but because the interaction energy was small compared to that between Ca^2+^- or V-sensors and channel opening. Sweet and Cox 2008 then tested a prediction that the interaction is bi-directional, by confirming that the apparent Kd for Ca^2+^-binding is reduced by altered by membrane depolarization. Furthermore, they concluded that the interaction occurred with RCK1 rather than RCK2, based on the effect of mutating the different Ca^2+^-binding sites. Finally, Savalli et al., 2012 used patch clamp fluorometry (on the VSD) to confirm that Ca^2+^ binding to the gating ring alters VSD activation. Thus, while the detailed approach used in the manuscript is new, neither the conclusions or even the use of patch clamp fluorometry appear to be novel. That said, the results could be valuable if they provide new insight into the nature of the interaction or its properties. But here too I have doubts. As the authors point out, the results show a strong correlation between effects on voltage-sensor activation and changes in the voltage-dependence of FRET. However, the results are described in very qualitative terms that don't seem to provide much insight beyond the basic conclusion. At minimum, I think the authors need to discuss whether the results are consistent or inconsistent with previous results, and make clear what is new.

One specific concern is whether the lack of effect of the γ1 subunit on FRET efficiency in Figure 2F is indeed consistent with the hypothesis that voltage-dependent FRET reflects VSD activation. The authors typically reference the effect of various mutations and regulatory subunits on the half-activation voltage of the VSD, V_h_(j). Strictly speaking, V_h_(j) is the half-activation voltage when channels are closed, and for most of the experiments may be an appropriate parameter to look at since voltage-dependent FRET is being measured under condition where a large fraction of channels are closed. However, in the case of γ1, voltage-dependent FRET is occurring over a range where channels are open (i.e. rel. conduct. In Figure 2F is saturated). While the authors are correct that Yan and Aldrich, 2010 proposed that V_h_(j) is unaltered by γ1, their model predicts a large shift in VSD half activation voltage when channels are open (because when channels are open, VSD activation depends on VSD-gate coupling, which is enhance by γ1). Therefore, is seems like one should expect γ1 to shift v-dependent FRET to more negative voltages if it was simply detecting VSD activation.

---

## [Author Response]

Essential revisions:1) Modify the TitleReviewers thought the title was vague and could be modified to better convey the message of the manuscript. The term "strongly coupled" also needs to be modified or justified since it implies a quantitative assessment of coupling that was not addressed.

We thank the reviewers for this suggestion. We have now changed the title to ‘Voltage-dependent dynamics of the BK channel cytosolic gating ring are coupled to the membrane-embedded voltage sensor’.

2) Address previous work and noveltyThe authors should provide a more complete comparison with previous work, especially with regards to VSD/gating-ring coupling or interaction, to help clarify "what is new" and whether the results are consistent or inconsistent with previous results.

We agree with the reviewers and we have added a paragraph to describe previous work and better clarify the novelty in our study:

‘Using the allosteric HA model of BK channel function, Horrigan and Aldrich (2002) proposed that Ca^2+^ binding to the Ca^2+^ bowl is coupled to the voltage sensor activation. […] This evidence indicates that the VSD is coupled to the gating ring, but none of these approaches directly monitored the conformational changes of the gating ring structure.’

3) Provide more guidance for interpreting the results:Indicating the expected status of voltage sensors under different conditions and manipulations would help readers to interpret the corresponding FRET data.

We are grateful to the reviewers for this recommendation. A new paragraph in has been included in the manuscript:

‘As discussed previously (Miranda et al., 2013), these changes in *E* with voltage are not conformational dynamics of the gating ring that simply follow the voltage dependence of VSD. […] However, we do not observe changes in *E* between 0 and +240 mV (Figure 1A). Similarly, at 100 µM Ca^2+^, charge movement takes place between -100 and +150 mV (Carrasquel-Ursulaez et al., 2015), while our FRET signals at 95 µM Ca^2+^ do not vary within this voltage range (Figure 1A).’

Provide more guidance for interpreting the results: Reviewers also desired more guidance on how to interpret the quantitative relationship between VSD activation and the voltage-dependence of FRET.

We agree with the reviewers in that a description of the quantitative relationship between VSD activation and voltage-dependence of FRET is lacking. Unfortunately, we have not found a satisfactory simultaneous fit of VSD activation (reflected by the voltage dependence of the macroscopic conductance) and the *E*-V data (see Miranda et al., 2013). One main limitation of our approach is the dynamic range of the *E-*[Ca^2+^] data. As in any FRET approach, information is restricted to a narrow window around the R_0_ value for the pair, which in our case is 50Å (see Miranda et al., 2013). In other words, any distance between fluorophores of >100 Å would result in a value of *E* close to 0, and similarly, any distance between the fluorophores shorter than 50 Å will give a value of *E* around 0.5 (this lower end distance is limited by the location of the fluorophore within the barrel of the protein). To provide data access to the readers, we have now generated a table with all values corresponding to the Boltzmann fits of G-V and *E*-V data, which has been added as Supplementary file 1. In addition, we are now indicating within the main text that there is a consistent correlation between the changes in the slope of the G-Vs data from S4 mutants with the changes in the steepness of the *E*-V relationship: ‘Interestingly, we also observe a correlation between the changes in the slope of the G-V curves and that of the *E*-V curves (see for instance, Figure 4F; see also Supplementary file 1)’.

Provide more guidance for interpreting the results: As well as several interesting or puzzling features of the data including the effect of Ba^2+^.

We agree with the observations of the reviewers and appreciate their suggestions regarding the effect of Ba^2+^. We have performed the experiments proposed by the reviewers, which are included in a new figure supplement (Figure 6—figure supplement 1). Our new results show that: 1) the *E*-V relations observed with Ba^2+^ and the S4 mutants (Figure 6) resemble the *E*-V relations with the RCK1 Ca^2+^ binding site knockout (D362A D367A) containing S4 mutations (BK667CY^D362A D367A R210E^ and BK667CY^D362A D367A E219R^). These observations confirm that the observed voltage-dependence of the Ba^2+^ is not related to the RCK1 site. These results are shown in Figure 6—figure supplement 1 and described in the revised manuscript version:

‘This conclusion is further supported by the lack of changes in Ba^2+^ responses when mutations in the VSD were made in a RCK1 Ca^2+^ binding site knockout (D362A D367A) background (Figure 6—figure supplement 1B and C.’

2) no shift in the voltage dependence of the FRET signal is observed after addition of the high affinity blocker bbTBA. Results are shown in Figure 6—figure supplement 1 and a new paragraph has been added to Discussion:

‘Why Ba^2+^ but not Ca^2+^? A possible answer for this question is that Ba^2+^ has the additional property of blocking the permeation pathway (Miller, 1987; Neyton and Miller, 1988; Zhou et al., 2012), which could somehow be transmitted allosterically to the gating ring. If simply ion permeation blockade is what matters, then we might expect that blocking permeation with the high affinity quaternary ammonium derivative *N*-(4-[benzoyl]benzyl)-*N,N,N*-tributylammonium (bb-TBA) (Tang et al., 2009) should produce a voltage dependent FRET signal with Ca^2+^ activation. But, it does not (Figure 6—figure supplement 1D). Another possibility for the Ba^2+^ effect could be a direct allosteric interaction between the intrinsic gating in the pore and the divalent binding site in RCK2, which needs to be tested further.’

Provide more guidance for interpreting the results: The remarkably steep Ca^2+^ dependence of FRET.

We thank the reviewers for this observation. To better appreciate the Ca^2+^-dependence of the FRET signal, we have looked at the *E* and G/Gmax data as a function of [Ca^2+^]. In Author response image 1, we have converted the data from Figure 1A-D to [Ca^2+^] dose-response curves. The *E*-[Ca^2+^] curves corresponding to the D362A/D367A and D362A/D367A 5D5A mutants are shown as average of all curves obtained at the various voltages. This representation evidences that the voltage effect on the *E* signal is observed within a specific range of Ca^2+^ concentrations (from around 4 to 50 μM). Within this range, the steepness of the *E*-[Ca^2+^] curves is similar to that of the G/Gmax-[Ca^2+^] data. We believe, as explained above, that the dynamic range of our *E-*[Ca^2+^] data is limited to a narrow window by technical constrains of the FRET technique.

Provide more guidance for interpreting the results: And the lack of voltage-dependence of FRET in 0 Ca^2+^.

As we have discussed above, we believe that the dynamic range of our *E-*[Ca^2+^] data is limited to a narrow window by technical constrains of the FRET technique. We have included the following in the manuscript: ‘As discussed previously (Miranda et al., 2013), these changes in *E* with voltage are not conformational dynamics of the gating ring that simply follow the voltage dependence of VSD. […] Similarly, at 100 µM Ca^2+^, charge movement takes place between -100 and +150 mV (Carrasquel-Ursulaez et al., 2015), while our FRET signals at 95 µM Ca^2+^ do not vary within this voltage range (Figure 1A).’

Provide more guidance for interpreting the results: Mechanistic or quantitative interpretation is lacking, and ideally should be improved, if only to provide some framework for future evaluation and tests. Proposing a model (i.e. a kinetic scheme) to account for the major features of the relation between Ca^2+^, voltage, and FRET efficiency (i.e. lack of FRET in 0 Ca^2+^ or high Ca^2+^, and correlations observed in the different mutants) would greatly strengthen the conclusions and enhance the significance of the results. While developing a model to account for all features of the data may be beyond the scope of this study, the authors should at minimum indicate which features of the data can or cannot be accounted for by previously proposed models (i.e. Miranda 2013) and whether their data provide any new insight into mechanism that must be included in gating models. This should be possible to do without incorporating additional experiments.

We agree with the reviewers that a model would provide some framework for future evaluation and tests. Unfortunately, we have been unsuccessful to adapt the HA model to describe the voltage dependencies of the *E* and G signals simultaneously. As described in Miranda et al., 2013, we explored a large number of modifications but all failed to describe the data. If we were to adopt a model, it should be reminiscent of, but simpler than, the one proposed by Savalli et al., 2012). In this model two different allosteric constants would account for the coupling of the gating ring to the PD, the VSD would only be allosterically coupled to the RCK1 (E’) and the RCK domains would be related by a new constant F. Notably, our data support E’ and C_1_’, but we have no information on C_2_’ or F. Given the premature nature of our understanding, we agree with the reviewers in that developing a model is beyond the scope of this study.

**Author response image 2. respfig2:** 

4) Reconcile discrepancies and apparent inconsistencies in the dataThe authors should address the apparent inconsistency between the effect of the γ1 subunit on FRET and its predicted effect on VSD activation.

We thank the reviewers for bringing up this apparent inconsistency. We agree that it was not adequately discussed in the manuscript. It is correct that, in the simplest scenario of the HA allosteric model, “a large shift in VSD half activation voltage when channels are open” (reviewer #3) would be expected. However, we would argue that a precise understanding of the allosteric coupling between the VSD and the pore, defined by D, should arise from the gating currents and their charge distributions and how they relate to the G-Vs and the presence or absence of γ1. We have had access to unpublished data, kindly provided by Carrasquel-Ursulaez and Latorre, showing gating currents, Q-V and G-V distributions in the presence and absence of gamma 1 subunit. Their findings support our statement “Analogously, no change in the voltage dependence of *E* was observed after co-expression of BKα with the γ1 subunit, which shifts the voltage dependence of pore opening by enhancing its allosteric coupling with the voltage sensor activation (Yan and Aldrich, 2010). As with the mutation F315A, the presence of γ1 subunit produces a minor shift in the Q-V distributions, not paralleling the large shift in the G-V curves (Carrasquel-Ursulaez and Ramon Latorre, personal communication).”, which has been included in the revised manuscript version.

As well as discrepancies between some of the results and data previously published.Results which seem to contradict earlier results include: a) GV shift magnitude of F315A in 100 Ca^2+^;

We agree with the reviewers that our results show smaller G-V shifts than those previously reported. The mutation F315A used in our study was introduced in the background of a human BK channel (GenBank accession no. U11058) construct, and is equivalent to the F380A mutant described in the Carrasquel-Ursulaez article, 2015. The differences in the observed G-V curves may be due, at least in part, to the use of different electrophysiological protocols and conditions than those described in Carrasquel-Ursulaez et al., specifically: 1) we use 300 ms long pulses instead of the 2 ms pulses used by Carrasquel-Ursulaez et al. 2) We use 40 mM symmetrical K^+^ rather than 110 mM K^+^ used by Carrasquel-Ursulaez et al. In addition, our construct has a fluorescent protein inserted. To what extent any of these differences contribute to a smaller shift in the GV, we do not know. We do not think this is a major issue, because we do observe large shifts (between ∆V_half_ of +100 to +200 mV, depending on Ca^2+^ concentrations) in the mutant, which are not accompanied by a shift in the *E*-V, and that is what it is important for this work. We have added a note in the legend of Figure 2 to make the reader aware of this apparent discrepancy: ‘It should be noted that the extent of the shifts induced by the mutation are smaller than previously reported (Carrasquel-Ursulaez et al., 2015), which could arise from the different experimental conditions and/or our fluorescent construct.’

b) Lack of a GV shift for the β3Nβ1 construct;

We are slightly puzzled by this comment. Data shown in Figure 3D indicate that β3Nβ1 induces a negative shift of the *E*-V curve. This is consistent with the Q-V data by Castillo et al., 2015; Figure 3 and Table S2, showing Q-V data for the different chimeras). To our knowledge, no G-V curves analysis data is shown in that paper. Why should we expect a shift in the G-Vs? Most amino acids of this chimera are from β3b, with just a few from the intracellular N-terminus of β1.

c) Whether they have adequate expression of β3b.

We thank the reviewers for this observation. First, we would like to state that we do have adequate expression of β3b. β3b RNA was always co-injected at molar ratios that were 25X larger that the BKa RNAs. This information is now provided in the Materials and methods section: ‘Auxiliary subunits (β3b, γ1 and chimera β3bNβ1) were co-injected with the BK667CFP/BK667YFP RNA mix at a 5:1 weight ratio, giving molar ratios above 20:1.’. To convince Dr, Lingle, we provide in Author response image 3 comparing the ionic currents obtained after co-expressing β3b plus non-fluorescently labelled BKα subunits (WT BKα) and β3b plus the BK667CY construct. Each panel showing ionic currents corresponds to a different construct indicated at the top. In each panel, current trace in black represents the ionic current in response to the most positive voltage step (+240 mV). These data show that when β3b is co-expressed with WT BKα we observe no difference with the effect described in the literature (e.g., Xia et al., 2000). Note that there is not an effect of blockade on the tail currents (see also G-Vs on the right, in yellow). Once β3b is co-expressed with the BK667CY construct, the kinetics of inactivation changed. It appears as if the off rate of inactivation is largely increased at positive potentials. Note also that there is a substantial current reduction at the tails as blockade increased with very positive potentials (see also G-V in blue). We use this behavior as an indication of β3b expression. The simplest interpretation is that the insertion of the fluorescent protein interferes with the kinetics of blockade mediated by the β3b NH2-terminal region (Lingle et al., 2001). Understanding this discrepancy will require further study. In any event, we consider this issue not affecting our conclusions. In this context, we agree that the main text needed to be modified to: ‘First, we co-expressed BK667CY α subunits with β3b and observed the expected inactivation of the ionic currents at positive potentials, yet with different blockade kinetics (see Figure 3—figure supplement 1) (Uebele et al., 2000; Xia et al., 2000; Lingle et al., 2001). […] The simplest interpretation is that the insertion of the fluorescent protein interferes with the kinetics of blockade mediated by the β3b NH2-terminal region (Lingle et al., 2001). Understanding this discrepancy will require further study.’

**Author response image 3. respfig3:**